# Structure and Properties of Sprayed Polyurethane Bio-Based Foams Produced Under Varying Fabrication Parameters

**DOI:** 10.3390/polym17182522

**Published:** 2025-09-18

**Authors:** Grzegorz Węgrzyk, Dominik Grzęda, Milena Leszczyńska, Laima Vēvere, Uģis Cābulis, Joanna Ryszkowska

**Affiliations:** 1Faculty of Materials Science and Engineering, Warsaw University of Technology, Wołoska 141, 02-507 Warsaw, Poland; dominik.grzeda.dokt@pw.edu.pl (D.G.); milena.leszczynska@pw.edu.pl (M.L.); joanna.ryszkowska@pw.edu.pl (J.R.); 2Polymer Laboratory, Latvian State Institute of Wood Chemistry, Dzerbenes 27, LV-1006 Riga, Latvia; laima.vevere@kki.lv (L.V.); ugis.cabulis@kki.lv (U.C.)

**Keywords:** spray-applied polyurethane foams, tall oil polyol, bio-based foams, processing parameters, mixing pressure, substrate temperature

## Abstract

The influence of both mixing pressure and substrate temperature on the structure and properties of spray polyurethane foams produced with a high content (80%) of tall oil-based polyol was investigated. The use of a renewable feedstock such as tall oil polyol aligns with the principles of sustainable development by reducing the carbon footprint and minimizing the environmental impact of the production process. The research focused on identifying the relationships between process parameters and the resulting materials’ thermal insulation properties, physico-mechanical performance, thermal behavior, cellular structure, and chemical composition. The results demonstrated that increasing the mixing pressure (from 12.5 to 17.5 MPa) and substrate temperature (from 40 to 55 °C) led to a reduction in average pore diameter, an increase in closed-cell content up to 94.5% and improved structural homogeneity. The thermal conductivity coefficient (λ) ranged from 18.55 to 22.30 mW·m^−1^·K^−1^ while apparent density varied between 44.0 and 45.5 kg·m^−3^. Higher mixing pressure positively affected compressive strength, whereas elevated substrate temperature reduced this parameter. Brittleness, water uptake, and dimensional stability remained at favorable levels and showed no significant correlation with processing conditions. These findings confirm the high quality of the materials and highlight their potential as sustainable, environmentally friendly insulation foams.

## 1. Introduction

In 2021, the global insulation materials market was valued at approximately USD 55 billion, with a compound annual growth rate of 5.4%. Among the insulation materials, polyurethane (PU) foams are widely used due to their excellent sealing, cushioning, and vibration-damping properties. The global revenue of the polyurethane foam insulation market was estimated at approximately USD 36,773 million in 2023. Various types of PU foams are employed—flexible, spray-applied, and rigid—across industries such as residential and non-residential construction, as well as the automotive sector [1]. Insulating materials constitute a significant category in the construction sector due to the increasing demand for energy savings in heating systems. Statistics concerning spray-applied polyurethane foam (SAPF) insulation [2] project that the market will reach USD 5.1 billion by 2032, with a projected compound annual growth rate of 9.5% over that period.

Spray-applied PU foam is particularly attractive for retrofitting and renovation applications in existing buildings, as it forms a continuous, monolithic barrier that provides outstanding thermal insulation and effectively minimizes heat loss. It is a versatile material suitable for insulation, roofing, and sealing thermal bridges.

With the growing adoption of circular economy policies, particularly in the construction sector, the demand for renewable raw materials is expected to rise significantly over the next five years. This will support the development and market position of polyurethane materials derived from renewable sources—especially tall oil-based polyols [3].

Another factor driving the growth of the insulation market is the search for solutions with higher energy efficiency [4]. Such solutions require changing the thermal resistance of the insulation, which depends, among other things, on its thickness and thermal conductivity. This characteristic of insulation depends, among other things, on the thermal conductivity of the solid matrix that forms the cell walls and struts of the foam [5,6]

The physico-mechanical and thermal insulation properties of polyurethane foams are significantly affected by factors such as the functionality and viscosity of the raw materials, the isocyanate-to-polyol molar ratio, and the concentration and type of blowing agents, catalysts, and other modifiers. These parameters, especially catalysts and blowing agents, strongly influence the foaming process and reaction temperature, leading to variations in cell structure and apparent density of the foams [6]. The final properties of polyurethane foams are thus highly process-dependent, as each stage of foaming involves specific chemical reactions.

Foam synthesis begins as soon as the components are mixed. Isocyanate groups react with hydroxyl groups, forming urethane linkages and causing a viscosity increase in the reaction mixture. Simultaneously, isocyanate groups react with water, producing urea derivatives and carbon dioxide, which expands the volume of the mixture. The isocyanate–water reaction is highly exothermic and can raise the core temperature of the foam block up to 165 °C. Exceeding this temperature may lead to thermal degradation or even spontaneous ignition of the material. Water is considered a reactive (chemical), environmentally friendly blowing agent [7,8].

The rate of foam formation is influenced by environmental conditions (ambient temperature, mold temperature, and component temperature), which in turn affect the cellular morphology and overall foam quality [9].

Lower foaming temperatures result in faster heat dissipation, reduced mass expansion, smaller cell sizes, and higher final densities [10]. At lower temperatures, polymerization occurs more slowly, reducing the overall reaction rate.

In addition to temperature, foaming process parameters such as mixing pressure [11,12], ambient temperature, and component temperature [13] also significantly impact PUR foam properties. Mixing speed has been identified as a critical variable [14].

According to the European Commission’s “A European strategy for plastics in a circular economy” [15], one of the key trends is to promote innovation and investment in closed-loop solutions, including the use of alternative raw materials to replace fossil-based resources. Among these are biobased polyols derived from vegetable oils, which have been used in polymer synthesis for decades. The first attempts to produce PUR foams from vegetable oils date back to the 1960s [16]. In recent years, attention has turned to polyols obtained from tall oil [16].

The production of PU foams requires specialized equipment, including high-performance mixing machines [11]. Spray-applied polyurethane foams are manufactured using the one-shot method, in which a two-component system—polyol blend and isocyanate—is mixed immediately prior to application. Upon mixing, urethane formation begins simultaneously with the isocyanate–water reaction, producing amines and CO_2_. These amines then react with residual isocyanates to form urea bonds. The evolving CO_2_ initially dissolves in the reactive mixture until saturation is reached. This is followed by nucleation, forming micro-bubbles—initiated, for example, by air introduced and dissolved during the mixing process. The amount of air incorporated depends on mixing parameters, which directly affect the number of pore nuclei formed.

The effects of mixing pressure and component temperature on the structure and properties of spray-applied polyurethane foams were investigated. The foams were synthesized using 80% tall oil polyol as a renewable component. Understanding these processing parameters is essential for elucidating the mechanisms of foam formation and their impact on the final material characteristics, including cell uniformity, mechanical performance, and thermal insulation efficiency. Optimizing pressure and temperature conditions during spray application is critical not only for achieving high-quality foams but also for ensuring their practical effectiveness. Furthermore, the use of tall oil polyol as an 80% renewable feedstock represents a significant step toward sustainable production by reducing the carbon footprint and environmental impact of the process.

## 2. Materials and Methods

### 2.1. Materials

#### 2.1.1. Polyurethane System

The subject of the research presented in this article is SAPF; the composition of this foam is described in Table 1. The name of the remaining ingredients (catalysts and surfactants) and the exact formulation of the foams constitute the know-how of the Polymer Laboratory, Latvian State Institute of Wood Chemistry, Riga, Latvia.

The foams were produced at an isocyanate index (INCO) of 110, where INCO is the ratio of the number of moles of the isocyanate component (NCO) used in the recipe to the number of moles of components containing hydroxyl groups (OH) and other groups capable of reacting with isocyanate groups.

#### 2.1.2. The Synthesis Process Information

The spray material was obtained by premixing component A according to the formula, after which it was applied using a spraying aggregate (GlasCraft, Lancs, UK) machine, model MH VR, equipped with a Probler P2 spray gun available at the Latvian State Institute of Wood Chemistry in Riga. At the ambient temperature of 22 °C and relative humidity of 27 & a single layer of foam was sprayed onto the substrate from a distance of 1 m. A series of spray trials was conducted to evaluate the influence of increasing mixing pressure (ranging from 12.5 to 17.5 MPa) and component temperature (ranging from 45 to 55 °C) on the synthesis process and the resulting foam properties.

An overview of the produced materials and corresponding process parameters is provided in Table 2.

The characteristic start time is the time from spraying to the moment of foam rising.

In Figure 1, a schematic description of the equipment is presented.

### 2.2. Methods

For each test condition, a representative sample of the rigid polyurethane foam layer—spray-applied onto an aluminum surface—was cut to a size of 0.5 m × 0.5 m and to a thickness typically used for closed-cell spray insulation.

From this initial foam section, test specimens of appropriate dimensions were subsequently extracted for further analyses, as illustrated schematically in Figure 2. Section A was used for the evaluation of thermal conductivity, apparent density, thermal stability, and water absorption. Sections B and C were prepared for mechanical strength testing, while Section D was allocated for the remaining characterization tests.

Sampling methodology

**Figure 2 polymers-17-02522-f002:**
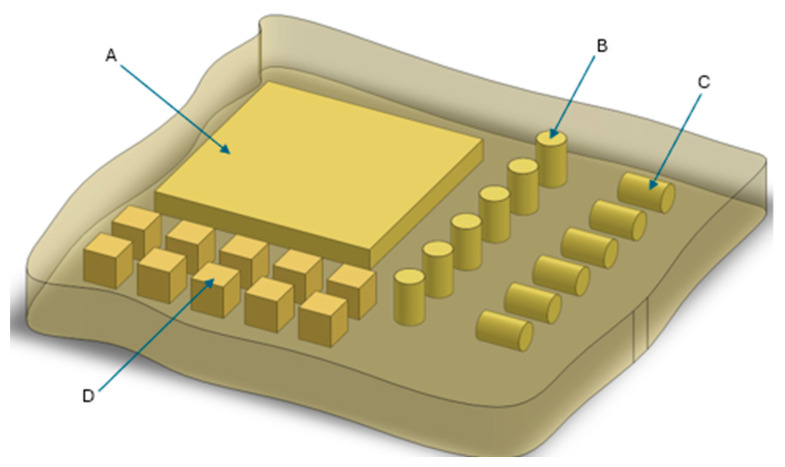
Schematic explanation of sample and specimen gathering. Samples: (**A**) Thermal coefficient, and stability (**B**) Parallel compression, (**C**) Perpendicular compression, (**D**) Density and other.

Scanning Electron Microscopy (SEM) and image analysis

Specimens were observed using a Hitachi TM3000 (Hitachi High-technology Corporation, Tokyo, Japan). Before observation, the samples were sprayed with a palladium–gold layer. Imaging was performed with secondary electrons at an acceleration voltage of 5 kV. Further image analysis was performed using ImageJ 1.54f software (National Institutes of Health, Bethesda, MD, USA). Based on the obtained images, cell density and cell size were determined. The detailed methodology of the SEM image analysis is provided in the Appendix A.

Thermogravimetric analysis (TGA)

Thermogravimetric analysis (TGA) was performed using a TGA Q500 instrument (TA Instruments, New Castle, DE, USA).

The samples were analyzed in an air atmosphere, with the temperature increased from ambient to 700 °C at a constant heating rate of 10 °C/min. The resulting data were processed using Universal Analysis 2000 software, version 4.5A (TA Instruments, New Castle, DE, USA).

Differential scanning calorimetry (DSC)

Differential scanning calorimetry (DSC) was performed to determine phase changes’ temperatures and thermal effects using a DSC Q1000 device (TA Instruments, New Castle, DE, USA). Foam samples were placed in hermetic aluminum pans, which were initially cooled down to −80 °C, heated to 160 °C at a rate of 10 °C/min (first heating cycle), cooled down again to −80 °C at a rate of 5 °C/min and reheated back to 160 °C at a rate of 10 °C/min (second heating cycle). The results were analyzed with Universal Analysis 2000 ver.4.5A software (TA Instruments, New Castle, DE, USA).

Fourier Transform Infrared Spectroscopy (FT-IR)

The chemical composition of the foams was analyzed based on absorption spectra obtained using a Nicolet 6700 spectrophotometer (Thermo Electron Corporation, Waltham, MA, USA) equipped with an attenuated total reflection (ATR) module.

Each sample was scanned 32 times in the wavenumber range of 4000–400 cm^−1^. The resulting spectra were processed using Omnic Spectra software, version 8.2.0 (Thermo Fisher Scientific Inc., Waltham, MA, USA).

The degree of phase separation (DPS) was determined according to the methodology described in [17].

Apparent density

The apparent density was determined in accordance with EN ISO 845 [18] by measuring the mass and volume of the samples. The sample mass was measured using an analytical balance with an accuracy of ±0.1 mg. The specimens had dimensions of 200 × 200 × 20 mm (cuboid) and their dimensions were measured with an accuracy of ±0.1 mm.

Physicomechanical properties

Friability, compressive strength, and closed-cell content of the composites were evaluated in accordance with ASTM C421, ISO 844 [19], and ISO 4590 [20], respectively.

Linear thermal stability tests were conducted according to ISO 2796:1986 [21], in the temperature range of −25 to 70 °C.

Water absorption was determined following the PN-EN ISO 29767 [22] standard.

Thermal Conductivity

The thermal conductivity coefficient (λ) of the foams was measured using a Linseis HFM (Linseis, Selb, Germany) 200 heat flow meter. The sample dimensions were 200 × 200 × 50 mm. The upper plate was maintained at 20 °C and the lower plate at 0 °C, in accordance with ISO 8301 [23].

Viscosity Testing

Viscosity measurements of the polyol blend were carried out using a Brookfield DV-II+ Pro viscometer (Brookfield Engineering Laboratories, Inc., Middleboro, MA, USA).

Surface temperature

Surface temperature of the sprayed polyurethane foam was measured using an infrared thermometer (Voltcraft IRF 260-10S (Conrad Electronic International, Hong Kong, China), optics 10:1, −50 to +260 °C, fixed emissivity 0.95). The pyrometer was positioned perpendicularly to the foam surface. Temperature readings were recorded starting from the moment of spraying at 1-s intervals for the first 120 s, followed by 5-s intervals until thermal stabilization was achieved.

## 3. Results and Discussion

### 3.1. Evaluation of Foaming Process Parameters for Polyurethane Foams

One of the key parameters in spray foam systems is the start time, defined as the reaction time from the mixing of the components to the onset of volume expansion.

Measurements of the start time revealed that mixing pressure had no significant effect on this parameter (Table 2). Increasing the mixing pressure of components A and B from 12.5 MPa to 17.5 MPa resulted in a slight change in the start time from 3.5 to 3.8 s for foams P125/T40 and P175/T40, respectively.

In contrast, increasing the temperature of the mixed substrates led to a significant reduction in the start time, from 3.8 to 1.0 s, for foams P175/T40 and P175/T55, respectively. This indicates a faster progression of chemical reactions and more rapid foam structure formation.

The recorded start times were shorter than those reported in the literature [24] and were sufficient to ensure proper application of the insulation layer via spray technique.

Both an increase in mixing pressure and an increase in component temperature contributed to a higher surface temperature of the foam, suggesting a rise in internal core temperature as well.

As noted by other researchers [11,13,25,26], the reaction temperature significantly influences the foaming and polymerization process.

### 3.2. Cellular Structure Analysis of the Foams

The foaming process resulted in polyurethane foams with structures presented in Figure 3 and Figure 4. Based on the attached SEM images, it can be concluded that cellular structure uniformity increases with higher mixing pressure (Figure 3) as well as with an increase in the temperature of the reaction mixture (Figure 4).

Cell size decreases, and the structure becomes more regular and homogeneous. It was also observed that the influence of the observation direction (parallel vs. perpendicular to the foam rise) becomes less pronounced at higher mixing pressures and elevated reagent temperatures.

Quantitative analysis of SEM images was performed. Mean pore diameter and cell density (NA) were determined in the directions parallel and perpendicular to the foam growth direction. The results of this analysis are presented in Table 3.

The average pore size analysis in sections taken parallel (dx) and perpendicular (dz) to the foam rise direction (Table 3) shows a clear trend: increasing the mixing pressure and reagent temperature leads to a decrease in the mean pore diameter of the foams.

In both orientations, an increase in cell density was observed with higher mixing pressures as well as with elevated reagent temperatures. At a mixing pressure of 12.5 MPa, an average of 46.7 cells·mm^−2^ (dx) and 49.5 cells·mm^−2^ (dz) was recorded. These values increased to 50.7 (dx) and 61.0 (dz) cells·mm^−2^ for foams produced at 17.5 MPa and 40 °C. Raising the reagent temperature to 55 °C further increased cell density to 66.1 (dx) and 68.0 (dz) cells·m^−2^.

Cell density in the perpendicular cross-section was consistently higher than in the parallel cross-section.

A higher cell density corresponds to a smaller average cell diameter, which may positively influence the thermal conductivity of the foams [27].

The results of the quantitative analysis of SEM foam images are presented in Figure 5 and Figure 6, where the distribution of cell density as a function of pore diameter is obtained through statistical image analysis.

In the transverse cross-sections of samples produced at 12.5, 15.5, and 17.5 MPa, the most frequently observed pore diameter was approximately 150 µm. In longitudinal sections, the dominant pore diameter was around 140 µm. The sample produced at 12.5 MPa showed a much broader pore size distribution compared to the others, indicating greater non-uniformity in cell size within that material.

The closed-cell content also increased from 89.6 to 93.5% with an increase in mixing pressure from 12.5 to 17.5 MPa. Increasing the temperature of the mixed substrates further improved the closed-cell content, reaching 94.5% for foam P175/T55—a value higher than that reported for commercial foams [28].

A comparison between pore size distribution results, planimetric analysis data, and closed-cell content suggests that higher mixing pressure limits pore opening.

The formation of open cells may result from the merging of multiple pores into larger ones, consistent with the mechanism described in [11].

Differences in pore size for foams produced at various mixing pressures may be attributed to changes in the number of microcell nuclei formed during the initial stage of the foaming reaction, resulting from increased mixing energy due to higher mixing pressure.

The observed differences in pore structure for foams produced at varying reagent temperatures (P175/T40–T55) may be explained by changes in the viscosity of the reactants. Therefore, the viscosity of component A was analyzed as a function of temperature. The results of this analysis are presented in Figure 7.

Viscosity measurements showed that as the temperature of component A increased, its viscosity decreased from 95 mPa·s at 35 °C to approximately 30 mPa·s at 60 °C. This trend is consistent with the typical behavior of Newtonian fluids.

Changes in component viscosity can significantly influence the formation of microcell nuclei—lower viscosity facilitates the nucleation of pores.

As the viscosity of the mixture decreases with increasing temperature, the mobility of reacting molecules increases, accelerating the polymerization rate. This correlation is supported by the measured reduction in foam start time.

Immediately after spray application, a dynamic polymerization reaction begins, leading to a rapid increase in system viscosity. This rise in viscosity contributes to greater foam stability during expansion. Higher foam stability—resulting from shorter start and gel times—limits mechanical deformation, such as gravitational sagging.

A shortened reaction time and faster gelation reduce the risk of pore rupture, as confirmed by the results on average pore diameters and closed-cell content for foams produced with increasing substrate temperatures.

The progression of the successive stages of cell morphology formation was presented, correlated with changes in temperature and viscosity over time. The course of the gelation process was discussed in connection with viscosity and temperature [29].

### 3.3. Thermal Analysis of the Foams

This study presents the results of foams produced using an identical chemical formulation, but under varying processing conditions.

Thermogravimetric analysis

To explain the observed differences in their structure, thermogravimetric analysis (TGA) was conducted. The results of this analysis are shown in Figure 8 and Figure 9.

Based on the mass loss curves (TG) shown in Figure 8 and Figure 9, the onset degradation temperature (Tonset) and the residual mass at 650 °C (P650) were determined.

From the derivative thermogravimetric (DTG) curves (Figure 7 and Figure 8), three distinct degradation stages were identified:
Stage 1: up to approximately 200 °C. In the first stage, the mass loss corresponds to the release of unbound, low-boiling-point components such as water, as well as the decomposition of polyols and isocyanates. In the tested materials, the degradation process begins during the first stage, where a peak corresponding to the maximum degradation rate Vmax1 is observed at temperature Tmax1.Stage 2: from 200 to 340 °C. In the second stage, degradation is mainly associated with the breakdown of both the hard and soft segments, with mass loss resulting from the cleavage of polyol and polyisocyanate bonds. Urethane bond scission in polyurethane polymers typically occurs in the range of 250–300 °C [30,31,32,33,34]. The second degradation stage is characterized by a single peak of maximum degradation rate Vmax2 at temperature Tmax2.Stage 3: from 340 to 650 °C. In the third stage, mass loss results from the degradation of aromatic products originating from the hard and soft segments, which are generated during the second stage. This phase also involves the decomposition of secondary and tertiary amines [31]. In the third stage, two distinct peaks are recorded, corresponding to maximum degradation rates Vmax3 and Vmax4, observed at temperatures Tmax3 and Tmax4, respectively.

The results of TG and DTG analysis for the foams are summarized in Table 4 and Table 5.

Based on the presented results, it can be concluded that the thermal characteristics of the foams vary with changes in mixing pressure at a constant substrate temperature of 40 °C.

Foam degradation begins at approximately 132 °C. As mixing pressure increases, the Tmax1 decreases, while the Vmax1 in this stage is favourably reduced by about 10% on average. These changes in Tmax1 and Vmax1 suggest that higher mixing pressures result in lower amounts of low-volatility substances and unreacted polyols and isocyanates remaining in the foam.

The Tmax2 decreases with increasing mixing pressure by up to 8 °C, while the maximum degradation rate at this stage (Vmax2) increases by approximately 12%. This indicates that higher mixing pressures lead to the formation of rigid segments with structures more susceptible to degradation at lower temperatures and higher rates.

The Tmax3 and Tmax4 increase with rising mixing pressure, as does the degradation rate (Vmax3 and Vmax4). Increase in degradation rates suggests that the temperature resistance of these foams is higher.

After degradation at 650 °C, the residual mass is lower in foams prepared at higher mixing pressures, indicating a reduction in the formation of polycyclic aromatic degradation products, as noted by [34].

Substrate temperature in the range of 40–55 °C, under constant mixing pressure of 17.5 MPa, does not significantly affect the temperature or rate of degradation in all degradation stages. This suggests that the phase structure of these samples is similar.

The amount of residue after degradation at 650 °C remains similar across all samples.

Differential Scanning Calorimetry analysis

The DSC thermograms are presented in Figure 10 and Figure 11. From the DSC curves, glass transition temperatures for the soft phase were identified as Tg1 = −28 °C and Tg2 = −19 °C (Table 6). These values remained unchanged across different processing conditions, indicating that the observed transitions are linked to the structure of the polyols used. For the tall oil-based polyol, a clear glass transition temperature Tgt was observed at approximately −24 °C (Appendix A). The pronounced Tg2 signal in the DSC curves of the tested foams likely corresponds to the soft segment containing the tall oil polyol, which constitutes approximately 80 wt.% of the polyols in components A.

Additionally, an endothermic peak corresponding to ordering transitions in the hard phase was observed in the DSC thermograms. This transition is characterized by the temperature transition (Tt) and enthalpy change of transition (ΔHt), with Tt ranging from 75.8 to 77.4 °C.

Increasing mixing pressure leads to higher transition enthalpy in the hard phase, indicating increased segment ordering. Conversely, increasing substrate temperature results in a tendency toward reduced transition enthalpy.

The relationship between the enthalpy change in the hard phase and the maximum degradation rate in the second degradation stage (Vmax2) was analyzed (Figure 12). This analysis confirms a correlation between hard phase ordering and the behavior observed during the second stage of thermal degradation.

Thermal conductivity

The results of the thermal conductivity of the foams are summarized in Table 7.

The thermal conductivity coefficient (λ) was within the range of 18.55–22.30 mW·m^−1^·K^−1^ and showed a slight increase with higher mixing pressure. Substrate temperature had no significant influence on the value of λ. This change may be attributed to differences in average pore diameter (Table 3), as well as to variations in the structure of the hard segment. An analysis of the relationship between the hard phase transition and insulation performance (Appendix A) suggests that increased enthalpy of the hard segment transition is associated with a higher thermal conductivity of the foams. Thermal insulation was also shown to correlate with the content of closed cells (Appendix A).

However, pore size variation in the range of 130–150 µm—as observed in the tested foams—did not significantly affect λ, which is consistent with the findings of Jarfelt and Rammas [35]. A reduction in pore size is generally beneficial for lowering thermal conductivity [36], and this correlation was confirmed in the tested foams. Substrate temperature had no significant influence on the value of λ.

The obtained thermal conductivity values are comparable to results for spray foams made with 1:1 biobased rapeseed polyol formulations, which exhibited λ values around 20 mW·m^−1^·K^−1^ [28]

Linear thermal stability

The results of the linear thermal stability of the foams are summarized in Table 7. The thermal stability was within the range of 0.17–0.53% and mixing pressure had no influence on the value of thermal stability of foams. There is a tendency for thermal stability to increase with increasing substrate temperature. It was observed that the error in measurement is significant.

### 3.4. Analysis of the Chemical Composition of the Foams

To confirm the presence of functional groups characteristic of polyurethane foams and to investigate structural differences resulting from varying foaming process parameters, FTIR analysis was conducted.

The results revealed absorption bands typical of polyurethane structures (Table 8 and Figure 13).

A broad absorption band in the wavenumber range of 3400–3200 cm^−1^ corresponds to the asymmetric and symmetric stretching vibrations of the –N–H group present in urethane and urea moieties.

The band with a maximum at 1510 cm^−1^ is attributed to the deformation vibrations of the same group. Bands with maxima at 2925 cm^−1^ and 2854–2852 cm^−1^ are associated with the asymmetric and symmetric stretching vibrations of –CH groups in CH_3_ and CH_2_ moieties.

Additional bands at 1453 cm^−1^ are due to asymmetric and symmetric bending vibrations of these groups, while the band at 1307 cm^−1^ is linked to stretching vibrations. The signal at 1705 cm^−1^ confirms the presence of carbonyl (C=O) groups in urethane and urea linkages. The peak at 1595 cm^−1^ is associated with the stretching vibrations of aromatic C=C bonds. The signal at 1411 cm^−1^ indicates the presence of trimerization products of isocyanates. The absorption band at 1217 cm^−1^ corresponds to C–N stretching vibrations, and the peaks at 1056–1055 cm^−1^ are attributed to C–O stretching vibrations, which are characteristic of the soft segments in polyurethane structures [37,38,39,40,41].

No significant shifts in the positions of the absorption bands were observed with changes in the foaming process parameters, indicating that the overall chemical structure of the materials remained consistent.

The rigid and flexible segments that make up polyurethane macromolecules are immiscible. This property allows for phase separation, leading to the formation of a soft and hard phase. The rigid segments that make up the hard phase often bond with each other through strong hydrogen bonds, which promotes phase separation. The phase separation process determines many physical and mechanical properties of polyurethanes, such as strength and thermal properties. Research on polyurethanes has demonstrated the ability to quantitatively assess the degree of phase separation (DPS) in these materials. This assessment is possible using various testing techniques, including DSC, TGA, and FTIR.

As part of this work, a DPS analysis using FTIR was performed based on the analysis of the multiplet bands in the 1600–1800 cm^−1^ range. This band corresponds to the stretching vibrations of the carbonyl group (C=O). A comparison of multiplet bands for foams produced at varying substrate mixing pressure is presented in Figure 13.

In the frequency range 1670–1760 cm^−1^, a number of bands resulting from the vibrations of the bonded and unbonded C=O groups occur (Figure 14). The degree of phase separation describes the contribution of rigid segments bonded to each other by hydrogen bonds. Quantitatively, DPS is calculated based on the half-bands constituting the multiplet band. First, the index of the hydrogen-bonded carbonyl groups of the urethane and urea bonds (R) is calculated as the ratio of the sum of the absorbance half-bands resulting from the vibrations of the hydrogen-bonded carbonyl groups of the urethane and urea bonds to the sum of the absorbance half-bands resulting from the vibrations of the unbonded carbonyl groups of the urethane and urea bonds. Then, DPS is calculated as the ratio of the carbonyl group index to the sum of the indices of these groups, increased by 1. A detailed description of DPS calculations is contained in many works, e.g., [42]. The results of the DPS analysis are summarized in Table 9.

FTIR spectral analysis revealed that the degree of phase separation (DPS) in the produced foams ranged from 0.49 to 0.64. An increase in the mixing pressure of the reactants led to a higher DPS, whereas for foams produced at 17.5 MPa, a decreasing trend in DPS was observed with increasing substrate temperature. Changes in DPS are associated with variations in the enthalpy of the hard phase transition (ΔHt) of the tested foams, as confirmed by the correlation analysis presented in the Appendix A.

### 3.5. Analysis of Physico-Mechanical Properties of the Foams

The results of the physico-mechanical property evaluation of the foams are summarized in Table 10.

Both changes in mixing pressure and substrate temperature affected the apparent density of the foams, which ranged from 44.0 to 45.5 kg·m^−3^. An increasing trend in apparent density was observed with rising temperature, likely due to the reduced initial viscosity of the raw materials [12].

The compressive strength measured in the direction parallel to the foam rise increased with mixing pressure, which may be attributed to improved homogeneity of the cellular structure. In the perpendicular direction, compressive strength increased when the mixing pressure was raised from 12.5 to 15.5 MPa.

At mixing pressure 17.5 MPa, a slight decrease in average compressive strength was observed. However, considering the standard deviation, it can be concluded that foam P175/T40 exhibited superior stability and structural uniformity.

An increase in substrate temperature resulted in lower compressive strength in both directions.

Compared to the similar foams [28], the compressive strength of the tested foams was significantly higher—both in relation to spray foams made with 40% rapeseed-based polyol and to commercial foams evaluated in that study.

Foam friability ranged from 0.4 to 0.7%, with the highest mass loss observed for sample P125.

Water absorption values ranged from 7.8 to 12.9 kg·m^−2^, with the highest value recorded for sample P175/T50.

For comparison, spray foams based on recycled PET polyols [43] exhibited water absorption values 2–3 times higher than those observed for the tall oil-based foams studied here.

## 4. Conclusions

This study presents a comprehensive analysis of the influence of mixing pressure and substrate temperature on the structure and properties of spray-applied polyurethane foams produced using 80% tall oil-based polyol.

An increase in mixing pressure from 12.5 to 17.5 MPa at a substrate temperature of 40 °C results in foams produced with these parameters:
a higher cell density and a corresponding reduction in average pore size,a significant increase in the closed-cell content was also observed,an increase in the degradation rate,a decrease in the amount of ash after degradation at 650 °C,an increase in the enthalpy of hard phase transformation,an increase in the thermal conductivity coefficient,an increase in the degree of phase separation,

In contrast, the following parameters remain unchanged: apparent density, compressive strength, friability, and water absorption.

An increase in substrate temperature from 40 to 55 °C at a mixing pressure of 17.5 MPa results in foams produced with these parameters:
a higher cell density and a corresponding reduction in average pore size,a significant increase in the closed-cell content was also observed,the degradation rate in subsequent stages of foam decomposition remains unchanged,the amount of ash after degradation at 650 °C remains unchanged,a tendency towards a decrease in the enthalpy of transformation in the hard phase is observed,the thermal conductivity coefficient remains unchanged,a decrease in linear thermal stability,a tendency towards a decrease in the degree of phase separation is observed,a not significantly decrease in apparent density,a decrease in compressive strength,a tendency towards a decrease in friability and water absorption is observed.

The produced bio-based foams are characterized by: the apparent density of the foams ranging from 44 to 45.5 kg·m^−3^, the thermal conductivity coefficient in the range of 18.55–22.30 mW·m^−1^·K^−1^, and the closed-cell content reaching up to 94.5%. These characteristics of the foams are comparable to those typically reported for commercial foams [44,45].

As part of the work, the relationships between various parameters describing the phase structure of foams were identified.

This work contributes meaningfully to the advancement of environmentally friendly polyurethane materials derived from renewable resources, aligning with global sustainability goals. The comprehensive evaluation of processing–property relationships provides a solid foundation for the future optimization of biobased polyurethane formulations.

## Figures and Tables

**Figure 1 polymers-17-02522-f001:**
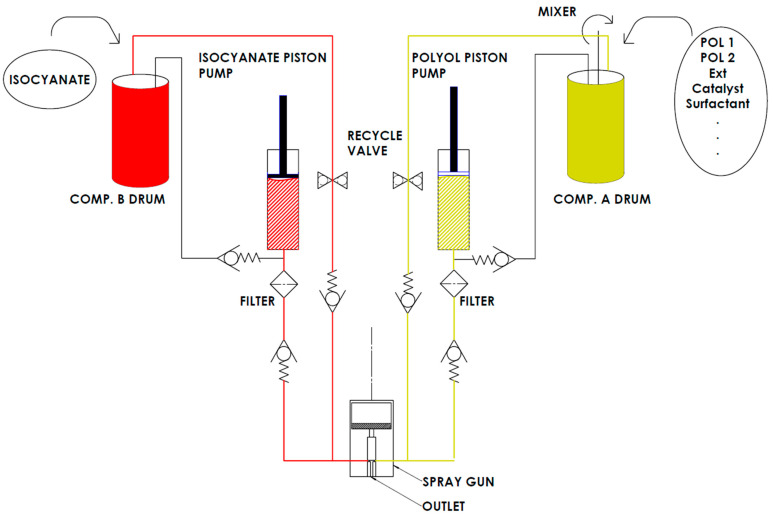
A scheme of synthesis equipment.

**Figure 3 polymers-17-02522-f003:**
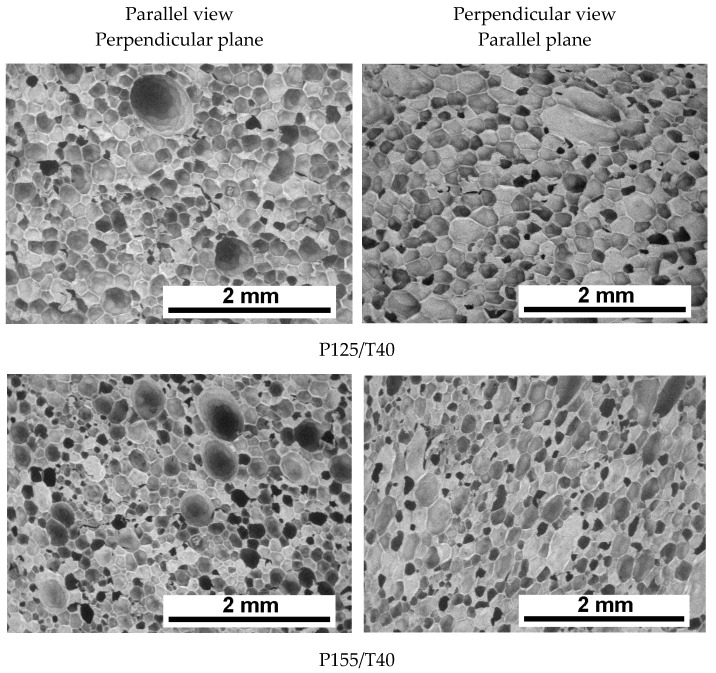
SEM micrographs of polyurethane foams obtained under varying mixing pressures.

**Figure 4 polymers-17-02522-f004:**
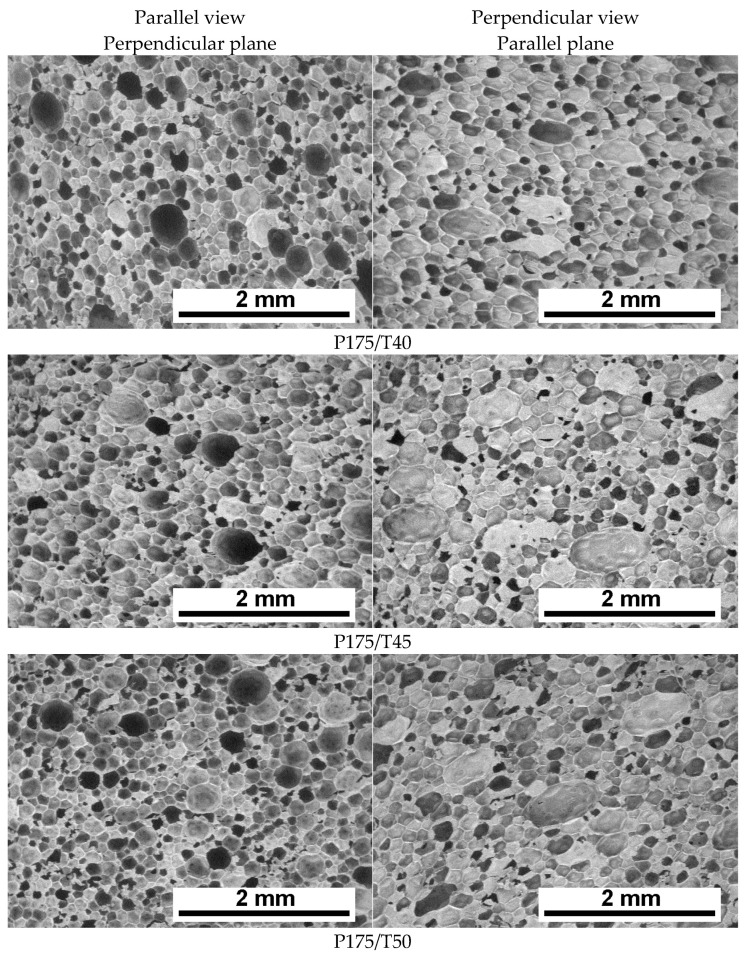
SEM micrographs of polyurethane foams obtained at various component temperatures.

**Figure 5 polymers-17-02522-f005:**
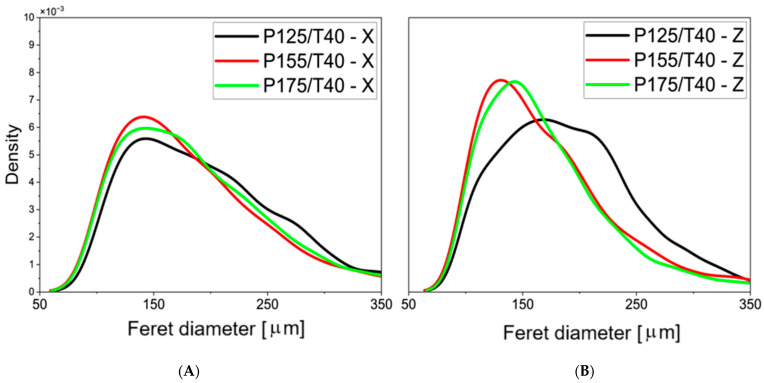
Statistical distribution of pore sizes determined using Feret diameter in the first part of the study; left panel (**A**): parallel to the foam rise direction; right panel (**B**): perpendicular to the foam rise direction.

**Figure 6 polymers-17-02522-f006:**
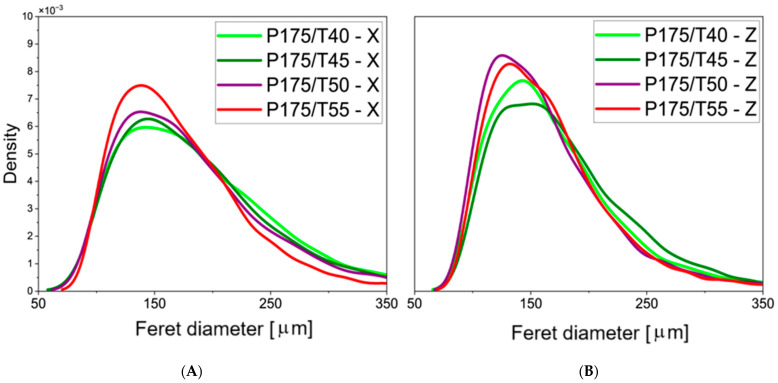
Statistical distribution of pore sizes determined using Feret diameter in the second part of the study; left panel (**A**): parallel to the foam rise direction; right panel (**B**): perpendicular to the foam rise direction.

**Figure 7 polymers-17-02522-f007:**
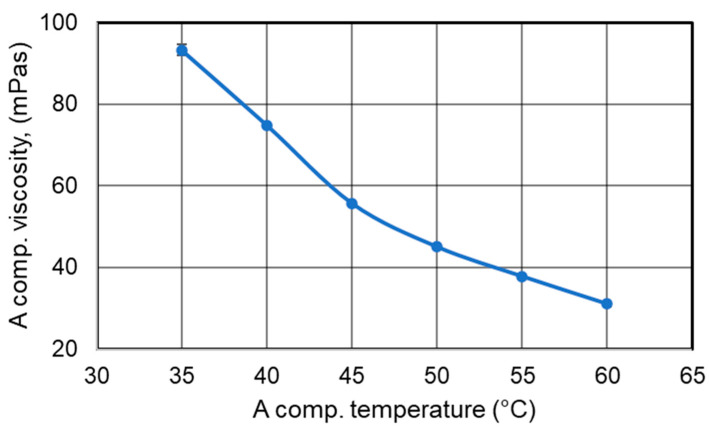
Change in Viscosity of Polyol Component A as a Function of Temperature.

**Figure 8 polymers-17-02522-f008:**
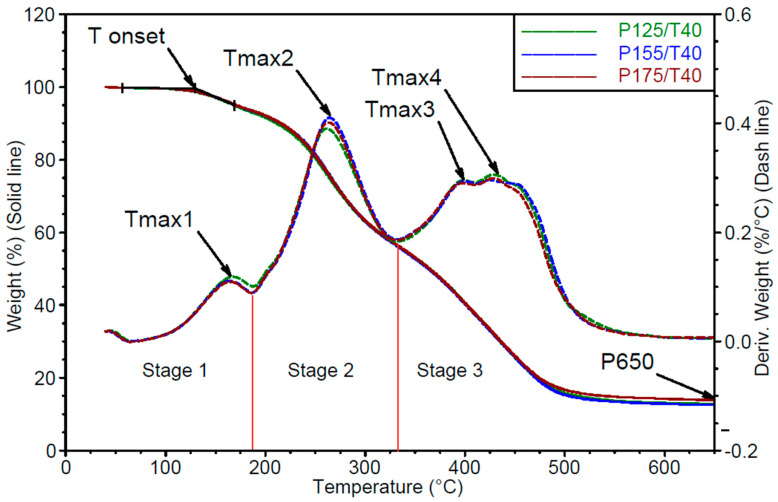
TG and DTG thermograms of samples from the series produced under different mixing pressures. Stages 1, 2 and 3 show following degradation steps.

**Figure 9 polymers-17-02522-f009:**
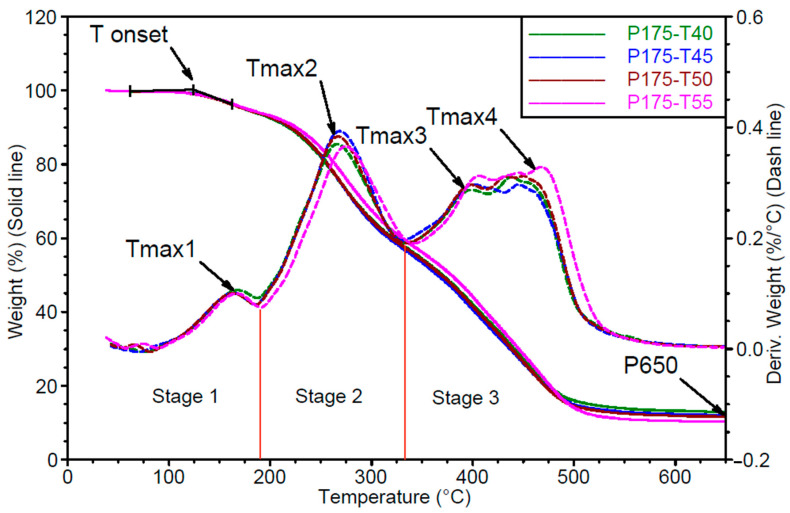
TG and DTG thermograms of samples from the series produced under different substrate temperatures. Stages 1, 2 and 3 show following degradation steps.

**Figure 10 polymers-17-02522-f010:**
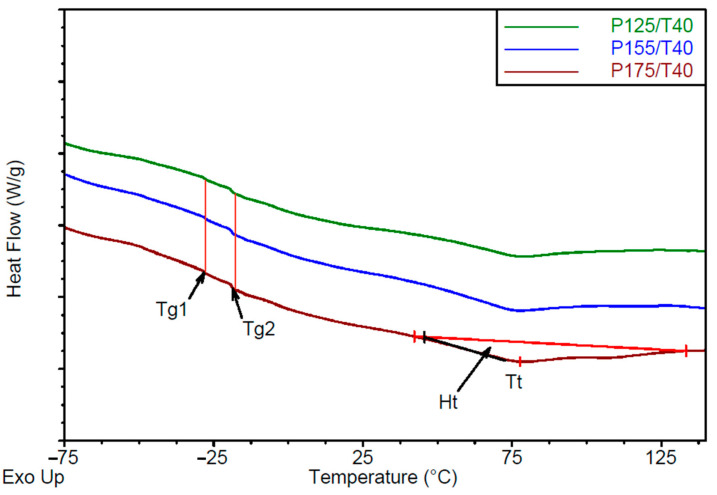
DSC thermograms of foams made at varying mixing pressures and a constant substrate temperature of 40 °C.

**Figure 11 polymers-17-02522-f011:**
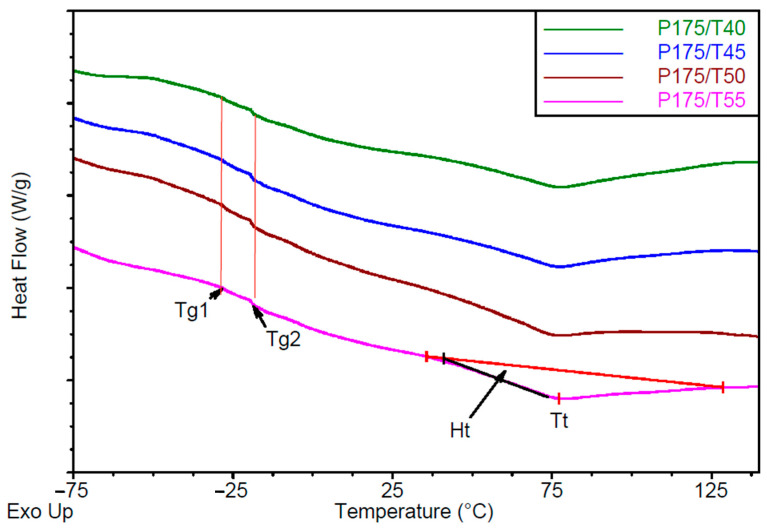
DSC thermograms of foams made at a constant mixing pressure of 17.5 MPa and varying substrate temperature.

**Figure 12 polymers-17-02522-f012:**
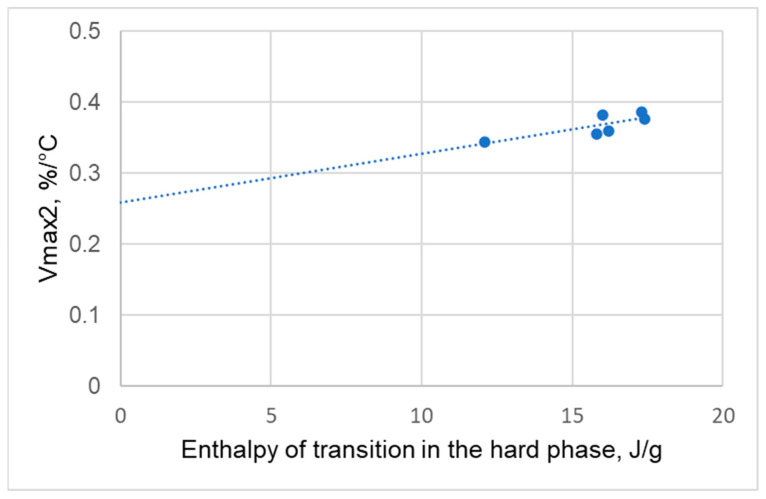
Relationship between hard phase transition enthalpy (ΔHt) and maximum degradation rate (Vmax2) in the second degradation stage.

**Figure 13 polymers-17-02522-f013:**
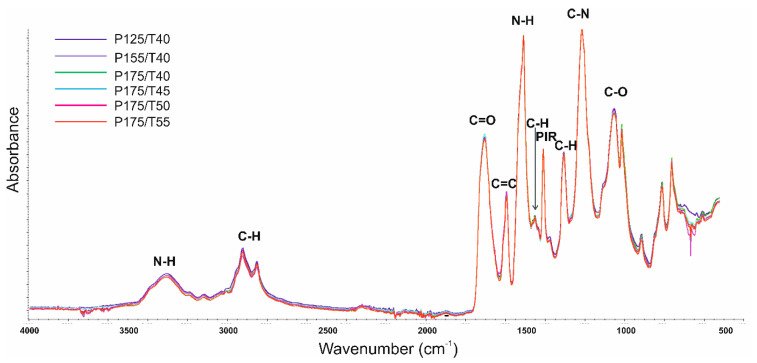
Comparison of FTIR spectra of foams prepared with different foaming parameters.

**Figure 14 polymers-17-02522-f014:**
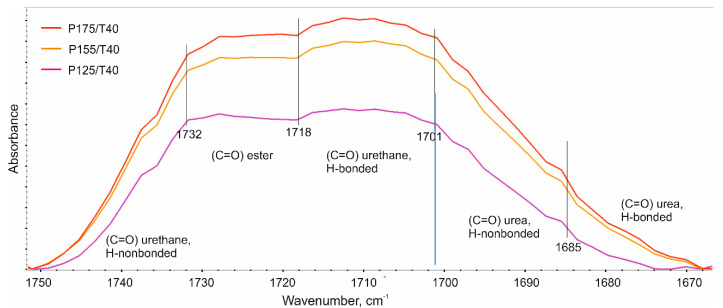
FTIR carbonyl region for polyurethane foams prepared at various mixing pressures with a constant substrate temperature of 40 °C.

**Table 1 polymers-17-02522-t001:** Formulation of rigid PU foams.

Components	Producer	Amount, wt. %
Component A		
Tail oil–based polyol (symbol TT)	Polylabs, Riga, Latvia	80
Polyol Lupranol 3300	BASF, Ludwigshafen, Germany	15
Diethylene glycol (DEG)	Sigma-Aldrich, St. Louis, MO, USA	5
Tris(1-chloro-2-propyl) phosphate (TCPP)–flame retardant	Albermarle (Louvain-la-Neuve, Belgium)	15
Water–blowing agent	PCC Rokita, Brzeg dolny, Poland	3.75
Catalysts	Evonik Industries AG, Essen, Germany	9.00
Surfactant	Evonik Industries AG, Essen, Germany	1.50
Component B		
MDI based isocyanate–pMDI	BASF, Ludwigshafen, Germany	159

**Table 2 polymers-17-02522-t002:** Description of Spray Process Parameters and Resulting Materials.

Process Parameters							
Sample name		P125/T40	P155/T40	P175/T40	P175/T45	P175/T50	P175/T55
Component A and B temperature, °C	A	40 ± 2	40 ± 2	40 ± 2	45 ± 2	50 ± 2	55 ± 2
	B	40 ± 2	40 ± 2	40 ± 2	45 ± 2	50 ± 2	55 ± 2
Component A and B pressure, MPa	A	12.5 ± 0.5	15.5 ± 0.5	17.5 ± 0.5	17.5 ± 0.5	17.5 ± 0.5	17.5 ± 0.5
	B	12.5 ± 0.5	15.5 ± 0.5	17.5 ± 0.5	17.5 ± 0.5	17.5 ± 0.5	17.5 ± 0.5
Hydraulic pressure, bar		40	50	60	60	60	60
Start time, s		3.5	3.5	3.8	2.3	1.2	1.0
Thickness of the foam, cm		3.3 ± 0.5	4.2 ± 0.7	3.0 ± 0.6	4.2 ± 0.5	4.5 ± 0.7	7.2 ± 0.9
Temperature on the surface of the foam, °C		101 ± 2	103 ± 2	107 ± 2	116 ± 2	118 ± 2	119 ± 2
**Spraying Aluminium Surface Parameters**							
Nominal thickness of surface material, mm		4 ± 0.1					
Spraying orientation, horizontal/vertical		Horizontal					
Surface temperature, °C		25 ± 2					

**Table 3 polymers-17-02522-t003:** Summary of Pore Structure Analysis Results of the Foams.

Mixing Pressure	Mean Pore Diameter, dx [μm]	Mean Pore Diameter, dz [μm]	Cell Density, dx [1·mm^−2^]	Cell Density, dz [1·mm^−2^]	Content of Closed Cells [%]
P125/T40	152 ± 58	149 ± 53	46.7	49.5	89.6 ± 1.9
P155/T40	142 ± 59	135 ± 57	52.1	57.6	90.3 ± 0.7
P175/T40	145 ± 59	133 ± 51	50.7	61.0	93.5 ± 0.3
P175/T45	145 ± 64	134 ± 49	49.5	60.4	93.3 ± 1.7
P175/T50	140 ± 56	126 ± 49	54.7	67.6	93.4 ± 1.6
P175/T55	129 ± 45	127 ± 46	66.1	68.0	94.5 ± 0.3

**Table 4 polymers-17-02522-t004:** Summary of TGA and DTG Analysis Results.

Sample	T_Onset_, °C	T_max1_, °C	V_max1_, %/°C	T_max2_, °C	V_max2_, %/°C
P125/T40	132 ± 1.2	173 ± 1.6	0.12 ± 0.001	289 ± 1.7	0.34 ± 0.018
P155/T40	132 ± 0.9	171 ± 1.2	0.10 ± 0.000	280 ± 2.2	0.36 ± 0.007
P175/T40	132 ± 2.9	169 ± 1.2	0.11 ± 0.001	271 ± 4.0	0.39 ± 0.020
P175/T45	125 ± 1.6	166 ± 1.7	0.10 ± 0.004	271 ± 3.1	0.38 ± 0.014
P175/T50	128 ± 0.9	166 ± 1.4	0.10 ± 0.001	270 ± 2.2	0.38 ± 0.003
P175/T55	130 ± 2.4	166 ± 1.7	0.10 ± 0.002	272 ± 1.2	0.36 ± 0.006

**Table 5 polymers-17-02522-t005:** Summary of TGA and DTG Analysis Results–continued.

Sample	T_max3_, °C	V_max3_, %/°C	T_max4_, °C	V_max4_, %/°C	P_650_, %
P125/T40	400 ± 3.3	0.19 ± 0.010	433 ± 3.6	0.19 ± 0.026	22.1 ± 1.5
P155/T40	397 ± 1.9	0.24 ± 0.021	455 ± 8.4	0.24 ± 0.051	18.0 ± 3.4
P175/T40	403 ± 2.8	0.30 ± 0.009	449 ± 9.5	0.32 ± 0.007	12.0 ± 0.8
P175/T45	403 ± 2.4	0.30 ± 0.002	454 ± 6.3	0.32 ± 0.013	12.0 ± 0.5
P175/T50	402 ± 3.1	0.30 ± 0.006	454 ± 4.2	0.32 ± 0.003	11.2 ± 1.0
P175/T55	403 ± 3.3	0.31 ± 0.005	459 ± 4.9	0.32 ± 0.009	12.4 ± 1.5

**Table 6 polymers-17-02522-t006:** Differential Scanning Calorimetry (DSC) Results.

Sample	Tg1 [°C]	Tg2 [°C]	Tt [°C]	DHt [Jg^−1^]
P125/T40	−28.1 ± 0.1	−19.0 ± 0.1	76.8 ± 0.3	12.1 ± 0.2
P155/T40	−28.1 ± 0.1	−19.2 ± 0.0	75.8 ± 0.1	15.8 ± 0.5
P175/T40	−28.1 ± 0.1	−19.0 ± 0.1	77.4 ± 0.1	17.3 ± 0.8
P175/T45	−28.1 ± 0.1	−19.0 ± 0.1	76.6 ± 0.1	17.4 ± 0.7
P175/T50	−27.9 ± 0.3	−19.0 ± 0.1	76.0 ± 0.7	16.0 ± 0.4
P175/T55	−28.0 ± 0.2	−19.0 ± 0.1	76.4 ± 1.0	16.2 ± 0.6

**Table 7 polymers-17-02522-t007:** Thermal conductivity and thermal stability of SFPU Foams.

Sample	λ [mW·m^−1^∙K^−1^]	Thermal Stability [%]
P125/T40	18.55 ± 0.02	0.51 ± 0.11
P155/T40	19.76 ± 0.02	0.32 ± 0.22
P175/T40	21.82 ± 0.03	0.53 ± 0.49
P175/T45	22.17 ± 0.09	0.45 ± 0.37
P175/T50	22.30 ± 0.06	0.17 ± 0.07
P175/T55	21.54 ± 0.08	0.25 ± 0.23

**Table 8 polymers-17-02522-t008:** Analysis of signal displacements in FT-IR spectroscopy of characterized materials.

P125/T40	P155/T40	P175/T40	P175/T45	P175/T50	P175/T55	
Wavenumbers [cm^−1^]						Bond (Vibration)
3305	3306	3304	3307	3307	3307	N-H (stretching)
2925	2925	2925	2925	2925	2925	C-H (asymmetric stretching)
2854	2853	2853	2852	2854	2854	C-H (symmetric stretching)
1705	1705	1705	1706	1705	1705	C=O (stretching)
1595	1595	1595	1595	1595	1595	C=C (stretching)
1510	1510	1510	1510	1510	1510	N-H (bending)
1453	1453	1453	1453	1453	1453	C-H (deformation)
1411	1411	1411	1411	1411	1411	PIR (deformation)
1307	1307	1307	1307	1307	1307	C-H (streching)
1217	1217	1217	1217	1217	1217	C-N (stretching)
1055	1056	1056	1056	1054	1056	C-O (stretching)

**Table 9 polymers-17-02522-t009:** Results of Phase Separation Degree (DPS) Analysis in Polyurethane Foams.

Sample	DPS
P125/T40	0.49 ± 0.5
P155/T40	0.60 ± 0.4
P175/T40	0.63 ± 0.6
P175/T45	0.64 ± 0.7
P175/T50	0.61 ± 0.4
P175/T55	0.62 ± 0.6

**Table 10 polymers-17-02522-t010:** Physico-Mechanical Properties of SFPU Foams.

Sample	Apparent Density [kg·m^−3^]	Compressive Strength		Friability [%]	Water Absorption, [kg·m^−2^]
		Para, kPa	Perp, kPa		
P125/T40	45.5 ± 1.43	248 ± 1.8	228 ± 9.4	0.68 ± 0.16	9.6 ± 0.3
P155/T40	45.1 ± 1.16	254 ± 6.7	232 ± 8.1	0.43 ± 0.14	12.1 ± 0.6
P175/T40	45.1 ± 0.56	255 ± 2.9	223 ± 2.1	0.53 ± 0.12	7.8 ± 0.8
P175/T45	45.0 ± 0.08	242 ± 7.6	202 ± 5.5	0.31 ± 0.03	11.4 ± 0.1
P175/T50	44.4 ± 0.67	251 ± 5.2	179 ± 2.5	0.37 ± 0.17	12.9 ± 0.3
P175/T55	44.0 ± 1.21	225 ± 8.9	165 ± 8.6	0.36 ± 0.23	8.9 ± 0.2

## Data Availability

The original contributions presented in this study are included in the article/Appendix A. Further inquiries can be directed to the corresponding author.

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
