# Peer review of "Structure and Properties of Sprayed Polyurethane Bio-Based Foams Produced Under Varying Fabrication Parameters"

_polymers, 2025, doi:10.3390/polym17182522_

Round 1

Reviewer 1 Report

Comments and Suggestions for Authors

Dear Authors,

The research objectives of the manuscript can be considered interesting, and the manuscript deserves consideration for eventual publication. However, the paper needs many revisions to satisfy science readers. The revision suggestions are attached. 

Author Response

Replies to the Reviewer's comments on the article “Structure and properties of sprayed polyurethane biobased foams produced under varying fabrication parameters”

We would like to thank the Reviewer for a thorough evaluation of our article and we believe that his observations will eliminate any shortcomings that may have appeared in the manuscript. Below we enclose the replies to the comments and recommendations made by the Reviewer. All of the changes in the text were highlighted.

Comment 1:  Abstract

The unit Bar should not be used.

44 and 45.5 kg/m3: 1) number of digits for these two values is different. 2) For unit writing

in the SI unit system, kg.m-3 is preferred; the same unit writing style should be applied to

all units. 

Response 1: We sincerely thank the Reviewer for their comments. In accordance with the Reviewer’s suggestion, the abstract has been revised.

The way of writing units has been changed throughout the text, the pressure unit has been changed from Bar to Pascal

Comment 2.

Introduction:  The introduction is generally very disorganized, it jumps from topic to topic. Many

paragraphs repeat each other. Some explanations are interspersed, then back to the

explanations in the previous paragraph.

  • Lines 37-48: These sentences are repetitive and too general. They should be Revised
  • lines 51-69: This part (if it is necessary, really) should be moved to the methods

section under the topic of thermal conductivity measurements, or can be given as

supplementary material. 

  • What is the aim of giving Table 1? (Table 1, if necessary, should be moved to the

experimental or results and discussion sections.)

  • In Table 1 and many others, instead of “decimal point”, “comma” was used.
  • For the pressure unit SI Unit Pascal should be used in abstract and in the whole

document. For one time in the experimental section the “bar

value“ correspondence of the Pascal values can be mentioned.

  • lines 70-76 and lines
  • lines 85-91 and lines 115-124: These explanations are well known for the PU

synthesis. It can be described shortly together with a chemical reaction

mechanism scheme in the PU synthesis section in the experimental section.

  • Through the introduction, specific studies and their findings should be mentioned

in whole sentences, including their important findings rather than giving the

references as numbers in parentheses. 

  • lines 103-114: these explanations are repeating; they can be shortened and

combined with the first paragraph(s) of the introduction.

Response 2: We sincerely thank the Reviewer for their comments. In accordance with the Reviewer’s suggestion, the introduction has been revised.

  • Lines 37-48: These sentences are repetitive and too general. They should be Revised

The text of these lines has been rephrased

  • lines 51-69: This part (if it is necessary, really) should be moved to the methods

section under the topic of thermal conductivity measurements, or can be given as

supplementary material. 

  • What is the aim of giving Table 1? (Table 1, if necessary, should be moved to the

experimental or results and discussion sections.)

Table 1 and its description have been removed from the introductory text.

  • In Table 1 and many others, instead of “decimal point”, “comma” was used.

In all tables and numbers, "comma" has been changed to "decimal point"

  • For the pressure unit SI Unit Pascal should be used in abstract and in the whole

document. For one time in the experimental section the “bar

value“ correspondence of the Pascal values can be mentioned.

Throughout the text, bar has been replaced with Pascal

  • lines 70-76 and lines 85-91 and lines 115-124: These explanations are well known for the PU

synthesis. It can be described shortly together with a chemical reaction

mechanism scheme in the PU synthesis section in the experimental section.

The text in lines 70-76, 85-91, 115 – 124 has been rephrased

  • Through the introduction, specific studies and their findings should be mentioned

in whole sentences, including their important findings rather than giving the

references as numbers in parentheses. 

These comments from the Reviewer have been followed throughout the introduction.

  • lines 103-114: these explanations are repeating; they can be shortened and

combined with the first paragraph(s) of the introduction.

The text in these lines has been rephrased

Most of the text in the introduction has been rephrased

Comment 3.

  1. Materials and methods

Lines 140: what is TT? 

lines 139-146: The material names are not given briefly. For the polyols used, technical

qualifications, such as type& chemical name , molar mass , viscosity, density, closed

formula etc should be given.

Table 2 Formulation of rigid PU foams: what is pbw?

In this table the function of each component can be given, rather than writing in long

sentences. 

In the paragraphs component A and component B are mentioned, however in Table 2 it is

not clear which is which?

“The foams were produced at an isocyanate index  (INCO) of 110 (see Table 2).” Table 2 is

not related with the above sentence. What is INCO abb?

2.1.2. The synthesis process information: This section is not clear enough. 

Table 3: Bar units should be converted into Pascal.

what does “start time” mean?

how did you measure surface temperature

for the thickness values from different parts of the PU foams there should be measurements

and the average values should be given with standart deviations.

for thickness of the plate the value was given as 4 mm, number of digits are different from

the other thickness values.

Ambient and surface parameters should be taken out of the table, can be given as short

wording.

For all the instruments in parentheses complete Company name, and city in addition to the

country should be given.

what is the reason to make DSC analysis between -80 to 160 oC? 

What does “The degree of phase separation (DPS)” mean? What is its importance? How it

was calculated should be explained shortly 

Thermal properties can be given as subtitles under the sub topic of thermal properties. It

may include 1) TG 2)DSC 3) Thermal stability ( note: what is this stability, do you mean

thermal cycling?) 4) Thermal conductivity.

Lines 243-244: What did they find briefly?

On the SEM images some dimensions of the cells can be labeled.

Tbale 4: What does NA mean? How did you calculate cell density? and what is cell density?

Figure 4 and 5: On the x axis what is this ferret?The figure components  can be numbered

as “a” and “b” in each of Figure 4 and Figure 5.

lies 298-300, lines 306-308 need reference support by also adding their findings.

what did reference 40 discuss?

TG discussion should be revised to clarify, the number of digits in paragraphs and in Tables

should be the same.

Table 5 and Table 6 should be combined and revised to be shortened. v values and Tmax

values can be shown on the TG-DTG figures, then unnecessary to put them into tables. The

mass loss percentages or the mass remaining percentage at the temperatures mentioned

can be given in the table

Figure 10 is not good enough. On the figures the real measured values should be marked,

rather than a lot of symbols. For Ht: It is the DeltaH value, and what is this value in number?

Heating and cooling cycles can be given on the same figure. 

Table 8 too confusing, unknown titles of the columns etc.

 Table 4: What does NA mean?

Cell density (NA)

How did you calculate cell density? and what is cell density?

Cell density is the average number of cells per square of surface area

Figure 4 and 5: On the x axis what is this ferret?

In Figures 4 and 5, the abbreviation ferret is used to describe the axes, instead of ferret diameter. This is the diameter of the object determined during stereological analysis.

The figure components  can be numbered  as “a” and “b” in each of Figure 4 and Figure 5.

lines 298-300, lines 306-308 need reference support by also adding their findings.

what did reference 40 discuss?

TG discussion should be revised to clarify, the number of digits in paragraphs and in Tables

should be the same.

Table 5 and Table 6 should be combined and revised to be shortened. v values and Tmax

values can be shown on the TG-DTG figures, then unnecessary to put them into tables. The

mass loss percentages or the mass remaining percentage at the temperatures mentioned

can be given in the table

Figure 10 is not good enough. On the figures the real measured values should be marked,

rather than a lot of symbols. For Ht: It is the DeltaH value, and what is this value in number?

Heating and cooling cycles can be given on the same figure. 

Table 8 too confusing, unknown titles of the columns etc.

Response 3: We sincerely thank the Reviewer for their comments. In accordance with the Reviewer’s suggestion, the materials and methods has been revised.

Lines 140: what is TT? 

Trade name of tall oil polyol

lines 139-146: The material names are not given briefly. For the polyols used, technical

qualifications, such as type& chemical name , molar mass , viscosity, density, closed

formula etc should be given.

Component A prepared by the Polymer Laboratory, Latvian State Institute of Wood Chemistry, Latvia, was used for the research conducted in this paper. The tallow polyol used as an ingredient in component A was manufactured by Polylabs, Riga, Latvia.

Due to the expertise of the Polymer Laboratory, Latvian State Institute of Wood Chemistry, Latvia, and the Polymer Laboratory, Latvian State Institute of Wood Chemistry, Latvia, we cannot disclose more detailed information about the foam formulation and ingredients than that presented in the text of the article.

Table 2 Formulation of rigid PU foams: what is pbw?

Instead of wt.% the abbreviation pbw was mistakenly used

In this table the function of each component can be given, rather than writing in long

sentences. 

In the paragraphs component A and component B are mentioned, however in Table 2 it is

not clear which is which?

In accordance with the Reviewer's instructions, the descriptions were changed and the contents of Table 2 were rephrased.

“The foams were produced at an isocyanate index  (INCO) of 110 (see Table 2).” Table 2 is

not related with the above sentence. What is INCO abb?

The foams were produced at an isocyanate index (INCO) of 110, where INCO is the ratio of the number of moles of the isocyanate component (NCO) used in the recipe to the number of moles of components containing hydroxyl groups (OH) and other groups capable of reacting with isocyanate groups.

2.1.2. The synthesis process information: This section is not clear enough.

The description of the synthesis process has been changed

Table 3: Bar units should be converted into Pascal.

Units changed from Bar to Pascal

what does “start time” mean?

The characteristic start time is the time from spraying to moment of foam rising

how did you measure surface temperature

Surface temperature of the sprayed polyurethane foam was measured using an infrared thermometer (Voltcraft IRF 260-10S (Conrad Electronic International, Hong Kong, China), optics 10:1, –50 to +260 °C, fixed emissivity 0.95). The pyrometer was mounted at perpendicular angle to the surface. Temperature was recorded from the moment of spraying at 1 s intervals for the first 120 s and then at 5 s intervals until thermal stabilization.

An addition was made to the article

for the thickness values from different parts of the PU foams there should be measurements

and the average values should be given with standart deviations.

for thickness of the plate the value was given as 4 mm, number of digits are different from

the other thickness values.

The data in Table 2 were supplemented and corrected.

Ambient and surface parameters should be taken out of the table, can be given as short

wording.

A change was made as suggested by the Reviewer.

For all the instruments in parentheses complete Company name, and city in addition to the

country should be given.

Dane o producentach zostały uzupełnione

What is the reason to make DSC analysis between -80 to 160 oC?

Within this temperature range, DSC analysis enables the complete characterization of the polyurethane. This characterization includes determining the glass transition temperature of the soft phase and characterizing the hard phase. Depending on the structure of the hard phase, its glass transition temperature can be determined and/or the enthalpy of transformation associated with the order change in this phase can be determined.

In the tested foams, the soft phase is formed by two types of polyols with glass transition temperatures below 0°C, while the hard phase transformation occurs in the range of approximately 50-140°C. Hence, the selected temperature range for testing.

What does “The degree of phase separation (DPS)” mean? What is its importance? How it

was calculated should be explained shortly 

Polyurethane macromolecules are composed of two types of segments: flexible segments and rigid segments. Flexible segments are made from polyols, while rigid segments are made from isocyanates and extenders and/or crosslinkers. Both types of segments are immiscible, differing in terms of their length and the level of intermolecular interactions. As a result of segment immiscibility, phase separation can occur, forming a soft phase and a hard phase. The rigid segments, which form the hard phase, are linked together by strong hydrogen bonds. Research on polyurethanes has shown that it is possible to assess the degree of phase separation (DPS) in these materials. This assessment is possible using various testing techniques: DSC, SAXS, WAXS, DMTA, TGA, and FTIR. Phase separation determines many properties of polyurethanes, such as strength and thermal properties.

As part of this work, DPS analysis was performed using FTIR. The DPS assessment is based on the analysis of multiplet bands in the range of 1600–1800 cm-1 corresponding to the stretching vibrations of the carbonyl group (C=O). Within the analyzed frequency range, a number of bands resulting from the vibrations of the bonded and unbonded C=O groups are present. The degree of phase separation describes the proportion of rigid segments interconnected by hydrogen bonds. Quantitatively, the degree of phase separation is calculated based on the half-bands comprising the multiplet band in the range of 1670–1760 cm-1. First, the index of the hydrogen-bonded carbonyl groups of the urethane and urea bonds (R) is calculated as the ratio of the sum of the absorbance half-values ​​resulting from the vibrations of the hydrogen-bonded carbonyl groups of the urethane and urea bonds to the sum of the absorbance half-values ​​resulting from the vibrations of the unbonded carbonyl groups of the urethane and urea bonds. Then, the DPS is calculated as the ratio of the carbonyl group index to the sum of the carbonyl group indexes increased by 1. A detailed description of the DPS calculations is contained in many works, e.g., Pretsch T. et al. [Pretsch T., Jacob I., Muller W.; Polym. Degrada. Stab. 2009, 94, 61.].

Thermal properties can be given as subtitles under the sub topic of thermal properties. It

may include 1) TG 2)DSC 3) Thermal stability ( note: what is this stability, do you mean

thermal cycling?) 4) Thermal conductivity.

Linear thermal stability of foams was assessed by the ISO 2796 standard, which is an assessment of the dimensional stability of foams that were exposed to elevated temperatures of 70°C and reduced temperatures (–20°C) for 72 hours.

Following the reviewer's suggestion, the structure of the text was changed

On the SEM images some dimensions of the cells can be labeled.

The inclusion of cell sizes in the figures makes image analysis difficult, so the scale is clearly marked in the figures.

Table 4: What does NA mean? How did you calculate cell density? and what is cell density?

Cell density (NA). Cell density is the average number of cells per square of surface area

Figure 4 and 5: On the x axis what is this ferret?The figure components  can be numbered

as “a” and “b” in each of Figure 4 and Figure 5.

In Figures 4 and 5, the abbreviation ferret is used to describe the axes, instead of ferret diameter. This is the diameter of the object determined during stereological analysis.

Corrected descriptions under drawings

lines 298-300, lines 306-308 need reference support by also adding their findings.

what did reference 40 discuss?

The text was corrected according to the reviewer's suggestion.

TG discussion should be revised to clarify, the number of digits in paragraphs and in Tables

should be the same.

Table 5 and Table 6 should be combined and revised to be shortened. v values and Tmax

values can be shown on the TG-DTG figures, then unnecessary to put them into tables. The

mass loss percentages or the mass remaining percentage at the temperatures mentioned

can be given in the table

The structure of the text describing the results of TG and DTG thermogram analysis has been changed. Numerical values ​​for Vmax and Tmax in subsequent stages of the degradation process have not been included in the figures. Each measurement was performed at least three times, the data in the graphs could be misleading. Unfortunately, figures with numerical values ​​became illegible. The readability of the data presented in Tables 5 and 6 has been improved.

Figure 10 is not good enough. On the figures the real measured values should be marked,

rather than a lot of symbols. For Ht: It is the DeltaH value, and what is this value in number?

Heating and cooling cycles can be given on the same figure. 

Each measurement was performed at least three times, the data in the graphs could be misleading. Unfortunately, figures  9 and 10 with numerical values ​​became illegible.

The delta Ht units are given in the table with the results of DSC thermogram analysis.

The figures only show DSC thermograms obtained in the first test cycle. After this test cycle, the phase structure of the PU changes. DSC analyses in the second heating cycle are conducted in our laboratories only to assess the accuracy of determining the glass transition temperature of the soft PU phase. We do not include these results in the figures with the results of DSC analysis of thermograms obtained in the second heating cycle, as they describe phenomena in PU with a different phase structure.

Table 8 too confusing, unknown titles of the columns etc.

Corrected column descriptions in table 8.

Comment 4.

Conclusions

The conclusion is too generalized. It seems that the statements and evaluations of the

results section are repeated in the conclusion. It lists various findings in words but does

not effectively summarize the overall significance of the study. 

There is a lack of structural integrity. 

The findings are presented in a fragmented and disorganized manner. 

No indication of the wider implications or applications of the findings. 

What is the important conclusion for researchers, engineers or designers? What do

researchers suggest for further studies?

The conclusion is not correlated with the introduction and aim of the study. 

In summary, it is recommended that the conclusion undergo several modifications to

enhance its overall structure and clarity. These changes might include reorganizing the

conclusion to make it more consistent, providing clearer information about the results, and

using a more scientific tone to present the findings in a more objective way.

Response 4: We sincerely thank the Reviewer for their comments. In accordance with the Reviewer’s suggestion, the conclusions has been revised.

Reviewer 2 Report

Comments and Suggestions for Authors

I believe the article can be published after the following minor adjustments:

lines 36-37: Vibration-damping properties were highlighted by the authors as one of the key qualities of PUR foams. It would have been nice to see a discussion of DMA characterization of these properties.

line 60 (Table 1): The authors should be consistent in using either dots or commas in indicating decimal place.   line 80: There seems to be a typo between "parameters" and "especially".   lines 510-524: Compressive strength was discussed in relation to pressure (also in lines 546-549). A stress-strain graph of the compression testing experiments could have helped in better understanding these parts.

Author Response

Replies to the Reviewer's comments on the article “Structure and properties of sprayed polyurethane biobased foams produced under varying fabrication parameters”

We would like to thank the Reviewer for a thorough evaluation of our article and we believe that his observations will eliminate any shortcomings that may have appeared in the manuscript. Below we enclose the replies to the comments and recommendations made by the Reviewer. All of the changes in the text were highlighted.

Comment 1: 

lines 36-37: Vibration-damping properties were highlighted by the authors as one of the key qualities of PUR foams. It would have been nice to see a discussion of DMA characterization of these properties.

Response 1: We sincerely thank the Reviewer for their comments. In accordance with the Reviewer’s suggestion, the abstract has been revised.

The way units are written has been changed throughout the text.

Dear Reviewer, this description shows the options that general PU can provide, however this study is mostly related to spraying rigid foam, where vibration-damping properties are not significant.

Comment 2: 

line 60 (Table 1): The authors should be consistent in using either dots or commas in indicating decimal place.  

Response 2: We sincerely thank the Reviewer for their comments.

Throughout the article, commas have been changed to dots.

Comment 3: 

line 80: There seems to be a typo between "parameters" and "especially".  

lines 510-524: Compressive strength was discussed in relation to pressure (also in lines 546-549). A stress-strain graph of the compression testing experiments could have helped in better understanding these parts.

Response 3: We sincerely thank the Reviewer for their comments.

Corrected as suggested, The stress-strain was added into supplementary material for your kind consideration.

The pressure changed the structure of the foam. Compression parameters are strongly related to structural distribution.

Reviewer 3 Report

Comments and Suggestions for Authors

Dear authors,

please check the following comments for improving your manuscript:

-"bio-based foam" instead of biofoams

-I'd suggest to replace the keywords, they are abstract

-"spray polyurethane foams", please rephrase, is spray a noun here?

-"cellular structure and chemical composition": never use a comma before the words "and" or "or" when for simple parathesis of similar things. The conjunction word already exists. Please check and correct where needed

-" 40°C" always keep a space between the numbers and units, check and correct in text where needed

-no need for abbreviation (CAGR), remove, use the whole phrase where needed

-I think PU is more correct than PUR

-I'd propose to remove the units from the equation, in text better

-table 1: decimals in English with periods instead of commas

-a space-line after table

-"The thermal conductivity of the solid matrix (i.e., cell walls and struts) and the gas phase within the cells can be influenced by modifying the chemical composition [18] [19] and the type of blowing agent used [18].": the cell geometry and density and the gas inclusion are of great importance in foams, please comment a bit more

-one dash in line 80

-"biobased polyols" is correct, bio-sourced as well

-SFP or sfPU or SFPUR...?

-"Polyurethane system": this is not a system, do you mean monomers? Reactants (the rest reagents included)?

-I suppose you know that a scheme mechanism would facilitate the reaction occurring understanding (described in Introduction)

-Please provide the chemical ingredients of Biobased polyol TT and Lupranol 3300. They are trade names, are their components known to you?

-Is H2O a chemical blowing agent or an initiator? Or a reagent for the production of CO2?

-font line 153

-regarding "spray" and "spraying" I would underline that spray is the transformation of a liquid to air cloud, due to temperature or pressure. In PU case, I would describe it as extrusion at RT... Consider and replace where needed in text

-"was sprayed onto the substrate from a distance of 1 meter." the oligomer had what diameter to be shaped, the ending hole?

-components A and B where mixed to 1:1 ratios in all cases, I assume

-the foams where tested after how many days/hours? Is then the polymerization considered practically done?

-any hold times in thermal methods applied (because the foam is insulating the energy transfer)? The calibration metals for DSC?

-viscosity measurements in which stage of the polymerization, after mixing of the components?

-I'd recommend to add the varying condition in captions of Fig. 2 and 3, so to facilitate the readers. 

-"planimetric analysis data", explain what that refer to?

-mm2, μm

-the difference in temperatures in viscosity results is not an issue?

-(why writting your results and comments in discussion in such short paragraphs? Why don't you unify the sentences in a single paragraph when referring to the same conclusion or chart?)

-for degradation rates I'd propose one decimal. The radical reactions occurring during pyrolysis are complicated, no great cosnlusion may occur by accuracy.

-we usually do not present st. dev. from thermal continuous characterizations. They are ok for the Tmax in your case, but not needed

-Tmax3 and Tmax4, correct in text where needed, correspondingly

Tg1

-what do you mean Tg1, Tg2. The material was post-polymerized in the instrument? What about the cross-linking? The amorphous polymer usually do not present Tg because of the not flexible macrochains, in your case?

-excellent IR for foams! They are taken by ATR at small flakes? They are all identical, as they should, since the same ingredients

-"Phase Separation Degree (DPS) Analysis", explain how is that performed and what does it reveal? Is fig. 13 trustful?

-g/m3, kg/m2

-after density results, the manufactured foam is a rigid foam, a medium one or a soft one?

-reshape the references by the Journal's guidelines

(I have not found the Supplementary Data)

Author Response

Replies to the Reviewer's comments on the article “Structure and properties of sprayed polyurethane bio-based foams produced under varying fabrication parameters”

We would like to thank the Reviewer for a thorough evaluation of our article and we believe that his observations will eliminate any shortcomings that may have appeared in the manuscript. Below we enclose the replies to the comments and recommendations made by the Reviewer. All of the changes in the text were highlighted.

Comments: bio-based foam" instead of biofoams

Response: Corrected as suggested

-I'd suggest to replace the keywords, they are abstract

Some keywords have been changed

-"spray polyurethane foams", please rephrase, is spray a noun here?

Corrected as suggested

-"cellular structure and chemical composition": never use a comma before the words "and" or "or" when for simple parathesis of similar things. The conjunction word already exists. Please check and correct where needed

Corrected as suggested

-" 40°C" always keep a space between the numbers and units, check and correct in text where needed

Corrected as suggested

-no need for abbreviation (CAGR), remove, use the whole phrase where needed

Corrected as suggested

-I think PU is more correct than PUR

Corrected as suggested

-I'd propose to remove the units from the equation, in text better

Rearranged

-table 1: decimals in English with periods instead of commas

Corrected as suggested

-a space-line after table

Corrected as suggested

-"The thermal conductivity of the solid matrix (i.e., cell walls and struts) and the gas phase within the cells can be influenced by modifying the chemical composition [18] [19] and the type of blowing agent used [18].": the cell geometry and density and the gas inclusion are of great importance in foams, please comment a bit more

Corrected as suggested

-one dash in line 80

Corrected as suggested

-"biobased polyols" is correct, bio-sourced as well

Corrected as suggested

-SFP or sfPU or SFPUR...?

SAPU

-"Polyurethane system": this is not a system, do you mean monomers? Reactants (the rest reagents included)?

We call it generally “system” as it is a mixture of different reactants and other ingredients that are mixed in one batch. That is why it is a name of paragraph

-I suppose you know that a scheme mechanism would facilitate the reaction occurring understanding (described in Introduction)

The introduction is changed

-Please provide the chemical ingredients of Biobased polyol TT and Lupranol 3300. They are trade names, are their components known to you?

Corrected

-Is H2O a chemical blowing agent or an initiator? Or a reagent for the production of CO2?

H2O a chemical blowing agent,  that is H2O is a reagent for CO2 production too. As a result of reaction H2O + -NCO -> carbamine -> primary amine + CO2 -> urea linkage.

It is either a chemical blowing agent and catalyst

-font line 153

Corrected

-regarding "spray" and "spraying" I would underline that spray is the transformation of a liquid to air cloud, due to temperature or pressure. In PU case, I would describe it as extrusion at RT... Consider and replace where needed in text

No, I disagree. Spraying is a perfect description to this type of application. Just before the spraying process, the substrates for creating PU are mixed in the mixing head

-"was sprayed onto the substrate from a distance of 1 meter." the oligomer had what diameter to be shaped, the ending hole?

The mixture of component A and component B, was sprayed onto the surfaces from a distance of 1 meter.

-components A and B where mixed to 1:1 ratios in all cases, I assume

Yes

-the foams where tested after how many days/hours? Is then the polymerization considered practically done?

After 14 days of seasoning, polymerization was considered to be practically complete.

-any hold times in thermal methods applied (because the foam is insulating the energy transfer)?

The calibration metals for DSC?

After cooling, the foam was kept for 1 minute and then the heating process began. For DSC calibration we use Indium

-viscosity measurements in which stage of the polymerization, after mixing of the components?

Viscosity measurements were done after mixing the component A (polyol, polyol, catalyst + …)

-I'd recommend to add the varying condition in captions of Fig. 2 and 3, so to facilitate the readers.

Descriptions have been added as suggested

-"planimetric analysis data", explain what that refer to?

Planimetric analysis is the image analysis method that measures the quantity of pores (or grains) per area.

-mm2, μm

Corrected

-the difference in temperatures in viscosity results is not an issue?

That is why we measured also the viscosity and described observations

-(why writting your results and comments in discussion in such short paragraphs? Why don't you unify the sentences in a single paragraph when referring to the same conclusion or chart?)

The descriptions have been rephrased as suggested

-for degradation rates I'd propose one decimal. The radical reactions occurring during pyrolysis are complicated, no great cosnlusion may occur by accuracy.

We agree with the comment that the data in the tables have been rounded. The processes occurring during pyrolysis are undoubtedly very complex. This work examined materials with the same chemical structure, but subjected to varying processing parameters, which resulted in changes in the phase structure of these materials. An attempt was made to analyze whether the differences in degradation rates could be linked to changes in the phase structure of the tested materials.

-we usually do not present st. dev. from thermal continuous characterizations. They are ok for the Tmax in your case, but not needed

We agree with the comment that the data in the tables have been changed

-Tmax3 and Tmax4, correct in text where needed, correspondingly

We agree with the comment that the data in the tables have been changed

Tg1

We agree with the comment that the data in the tables have been changed

-what do you mean Tg1, Tg2. The material was post-polymerized in the instrument? What about the cross-linking? The amorphous polymer usually do not present Tg because of the not flexible macrochains, in your case?

This article presents the results of Tg1 and Tg2 analyses determined during the first heating cycle. Samples were cooled to -80°C before heating.

Polyurethanes are two-phase materials composed of a soft phase formed by flexible segments derived from polyols and a hard phase formed by rigid segments derived from isocyanates and extenders and/or crosslinkers. Depending on the content of rigid segments in the PU, polyurethanes exhibit the characteristics of either a flexible or rigid material. However, they remain a two-phase material. In the tested materials, Tg1 and Tg2 refer to the soft phase of PU composed of two types of polyols, therefore, the soft phase of these materials is described by two glass transition temperatures.

After the first heating process, post-polymerization can occur. This is when the phase structure of the PU typically changes, which is why we only analyze the first heating cycle in our PU studies.

-excellent IR for foams! They are taken by ATR at small flakes? They are all identical, as they should, since the same ingredients

The tested materials have identical chemical structures, but different phase structures. Therefore, similar bands are visible in the spectra. Only the intensity of the bands changes, which allows the phase structure of the foam material to be analyzed.

-"Phase Separation Degree (DPS) Analysis", explain how is that performed and what does it reveal? Is fig. 13 trustful?

Polyurethane macromolecules are composed of two types of segments: flexible segments and rigid segments. Flexible segments are made from polyols, while rigid segments are made from isocyanates and extenders and/or crosslinkers. Both types of segments are immiscible, differing in terms of their length and the level of intermolecular interactions. As a result of segment immiscibility, phase separation can occur, forming a soft phase and a hard phase. The rigid segments, which form the hard phase, are linked together by strong hydrogen bonds. Research on polyurethanes has shown that it is possible to assess the degree of phase separation (DPS) in these materials. This assessment is possible using various testing techniques: DSC, SAXS, WAXS, DMTA, TGA, and FTIR. Phase separation determines many properties of polyurethanes, such as strength and thermal properties.

As part of this work, DPS analysis was performed using FTIR. The DPS assessment is based on the analysis of multiplet bands in the range of 1600–1800 cm-1 corresponding to the stretching vibrations of the carbonyl group (C=O). Within the analyzed frequency range, a number of bands resulting from the vibrations of the bonded and unbonded C=O groups are present. The degree of phase separation describes the proportion of rigid segments interconnected by hydrogen bonds. Quantitatively, the degree of phase separation is calculated based on the half-bands comprising the multiplet band in the range of 1670–1760 cm-1. First, the index of the hydrogen-bonded carbonyl groups of the urethane and urea bonds (R) is calculated as the ratio of the sum of the absorbance half-values ​​resulting from the vibrations of the hydrogen-bonded carbonyl groups of the urethane and urea bonds to the sum of the absorbance half-values ​​resulting from the vibrations of the unbonded carbonyl groups of the urethane and urea bonds. Then, the DPS is calculated as the ratio of the carbonyl group index to the sum of the carbonyl group indexes increased by 1. A detailed description of the DPS calculations is contained in many works, e.g., Pretsch T. et al. [Pretsch T., Jacob I., Muller W.; Polym. Degrada. Stab. 2009, 94, 61.].

We have been conducting DPS analysis of polyurethanes for many years. We conducted research with chemists from our university, who were initially skeptical about the feasibility of this analysis. We analyzed their materials. We performed FTIR analyses several times for each material, and each spectra was analyzed. We obtained reproducible analytical results. Calculating DPS requires the development of a repeatable spectral analysis procedure, which ensures fully reliable results. These analytical procedures have been developed and are used in our research.

-g/m3, kg/m2

corrected

-after density results, the manufactured foam is a rigid foam, a medium one or a soft one?

The type of foam is not determined by density, but by its strength characteristics, e.g. hardness.

-reshape the references by the Journal's guidelines

The list of publications has been corrected

Round 2

Reviewer 1 Report

Comments and Suggestions for Authors

Dear Authors,

The revised manuscript is clear and can satisfy science readers. 

The list of references is repeated two times.

Reviewer 3 Report

Comments and Suggestions for Authors

Please resize the figures, in some cases are too big, and the tables, in some cases they are splitted